# OS-ATLAS: A FOUNDATION ACTION MODEL FOR GENERALIST GUI AGENTS

**Zhiyong Wu**[1]*, **Zhenyu Wu**[1,2]*, **Fangzhi Xu**[1]*, **Yian Wang**[2]*, **Qiushi Sun**[3], **Chengyou Jia**[1], **Kanzhi Cheng**[1], **Zichen Ding**[1], **Liheng Chen**[3], **Paul Pu Liang**[4], **Yu Qiao**[1]

[1]Shanghai AI Laboratory [2]Shanghai Jiaotong University
[3]The University of Hong Kong [4]MIT
wuzhiyong@pjlab.org.cn
https://osatlas.github.io/

## ABSTRACT

Existing efforts in building GUI agents heavily rely on the availability of robust commercial Vision-Language Models (VLMs) such as GPT-4o and GeminiPro-Vision. Practitioners are often reluctant to use open-source VLMs due to their significant performance lag compared to their closed-source counterparts, particularly in GUI grounding and Out-Of-Distribution (OOD) scenarios. To facilitate future research in this area, we developed OS-Atlas —a foundational GUI action model that excels at GUI grounding and OOD agentic tasks through innovations in both data and modeling. We have invested significant engineering effort in developing an open-source toolkit for synthesizing GUI grounding data across multiple platforms, including Windows, Linux, MacOS, Android, and the web. Leveraging this toolkit, we are releasing the largest open-source cross-platform GUI grounding corpus to date, which contains over 13 million GUI elements. This dataset, combined with innovations in model training, provides a solid foundation for OS-Atlas to understand GUI screenshots and generalize to unseen interfaces. Through extensive evaluation across six benchmarks spanning three different platforms (mobile, desktop, and web), OS-Atlas demonstrates significant performance improvements over previous state-of-the-art models. Our evaluation also uncovers valuable insights into continuously improving and scaling the agentic capabilities of open-source VLMs.

## 1 INTRODUCTION

With the recent adoption of large language models (LLMs), the fantasy of building digital agents (Wu et al., 2024)—similar to *JARVIS* in The Iron Man—to automate daily tasks is evolving from science fiction into a tangible reality. Many current agents make decisions based on textual descriptions of the environments, such as HTML and accessibility trees, which is often lengthy (Zheng et al., 2024a), noisy (Cheng et al., 2024; WebAIM, 2024), and hard to acquire in practice. More recent studies (Cheng et al., 2024; Hong et al., 2024b; Li et al., 2024) have explored the use of large vision-language models (VLMs) to develop graphical user interfaces (GUI) agents capable of performing complex tasks simply by analyzing the screen - an information-complete medium for agent's decision-making, allowing for greater flexibility. At the core of a GUI agent lies an **action model** that enables **GUI grounding** - the process of transforming natural language instructions into executable actions within the operating system (e.g., clicking somewhere on the screen).

Despite their advancements, existing open-source VLM-based GUI action models have been criticized for their poor performance in GUI grounding and generalizing to Out-Of-Distribution (OOD) scenarios (Lu et al., 2024b; Chai et al., 2024), significantly restricting their applicability in real-world situations. The ineffectiveness of current models can be attributed to two primary factors.

First, most existing VLMs are rarely pretrained on GUI screenshot images. While some early efforts have focused on gathering screenshots corpus for websites (Lee et al., 2022; Chen et al., 2024b)

---

* Equal Contribution.

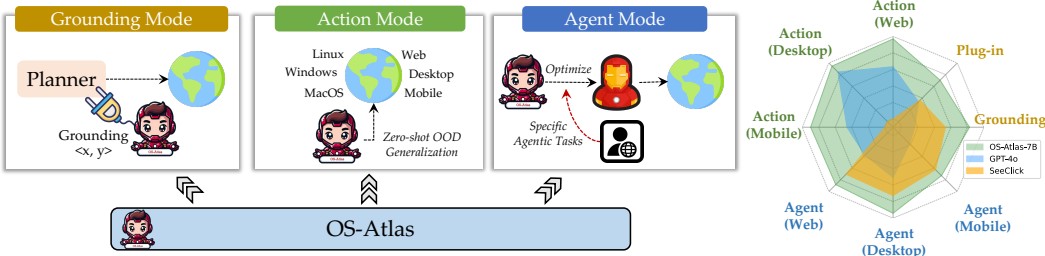

Figure 1: (Left) The OS-Atlas model operates in three distinct modes to cater to various research needs. In Grounding mode, OS-Atlas predicts element coordinates based on user instructions and can be integrated with a planner module to create a complete agent. In Action mode, OS-Atlas functions independently to solve step-level agent tasks universally across different platforms and applications, even in zero-shot OOD scenarios. In Agent mode, OS-Atlas undergoes further supervised fine-tuning to address specific agent tasks. (Right) Overall performance comparisons between OS-Atlas and other state-of-the-art models.

and mobile applications (He et al., 2020; Wang et al., 2021), there remains a significant lack of a large-scale, open-source corpus of screenshots that encompasses multiple platforms (Windows, MacOS, Linux, iOS, Android), a variety of applications, and different resolution sizes. Given that all GUIs operate under similar design principles, we believe that pre-training on such a comprehensive corpus would enable GUI agents to achieve better GUI grounding, especially in OOD generalization.

Second, the heterogeneity of content and format in existing datasets (Zhang et al., 2024c; Chen et al., 2024c), along with the issue of action naming conflicts, further undermines generalization. In current datasets, the same action is often labeled with different names across platforms. For instance, the "tap" action on mobile devices and the "click" action on desktop platforms are logically equivalent yet labeled differently. This inconsistency can create confusion during model training and ultimately result in decreased performance.

In this work, we are motivated to build a strong foundation action model to facilitate the development of future generalist GUI agents. Toward this goal, we make the following contributions:

1. We have developed and released the first multi-platform GUI grounding data synthesis toolkit. This toolkit enables the automatic synthesis of GUI grounding data across various platforms, including Windows, macOS, Linux, Android, and the Web. By doing so, it significantly reduces the engineering efforts required for data curation in future research.

2. Leveraging this data toolkit, we curated and open-sourced the largest multi-platform GUI grounding corpus to date, which comprises over 2.3 million distinct screenshots and more than 13 million GUI elements. Notably, our corpus includes desktop grounding data that has not been present in previous works. To facilitate evaluation of GUI grounding, we identify and re-annotate 11.32% incorrect samples in the popular benchmark ScreenSpot (Cheng et al., 2024) and release ScreenSpot-V2.

3. Through the above data innovation and an approach to resolving action naming conflicts during training, we developed OS-Atlas, a highly accurate foundation action model that operates universally across all GUIs. OS-Atlas can function in three different modes when developing GUI agents as depicted in Figure 1.

4. We present the most comprehensive evaluation of GUI agents to date, covering six benchmarks across three different platforms: desktop, mobile, and web. As shown in Figure 1, OS-Atlas demonstrates a superior performance improvement over previous SOTA models. This strong performance indicates that OS-Atlas can serve as an open-source alternative to powerful commercial VLMs, such as GPT-4o, for developing future GUI agents.

## 2 RELATED WORK

**GUI Agents and Large Action Models.** Autonomous agents powered by LLMs, known as language agents (Weng, 2023; Sumers et al., 2023), have recently garnered significant attention due

to their interactive capabilities (Wang et al., 2023; Sun et al., 2023; Hong et al., 2024a; Durante et al., 2024). Recent efforts have begun to enable agents to interact with operating systems via programs (Sun et al., 2024) or API calls (Wu et al., 2024; Zhang et al., 2024a). However, the closed-source nature of most commercial software imposed significant limitations, as agents don't have access to their internal APIs or codes. Consequently, research shifts toward GUI-based agents that interact with digital devices through human-like mouse and keyboard actions (Cheng et al., 2024; Hong et al., 2024b; Zheng et al., 2024a). To facilitate effective agent interactions, Large Action Models (LAMs) have been developed to address general agentic tasks by interpreting human intentions and predicting actions in the form of function-calling (Zhang et al., 2024c;b; Zeng et al., 2023; Yin et al., 2023). Nevertheless, progress is hindered by the limited quantity and vast diversity of available agent data (Li et al., 2024; Xu et al., 2024). Specifically, LAMs focusing on GUI interactions remain underexplored, with only a few attempts made to train GUI grounding models or agents (Cheng et al., 2024; Hong et al., 2024b; Gou et al., 2024).

To the best of our knowledge, OS-Atlas is the first LAM specifically designed for GUI agents.

**GUI Executable Language Grounding.** The core functionality of an LAM is to convert natural language (NL) instructions into actions and associated parameters (e.g., element coordinates), commonly known as GUI Executable Language Grounding, or simply GUI grounding. Existing GUI grounding training data can be divided into two types: referring expression grounding (REG) (Liu et al., 2023) and instruction grounding (IG) (Li et al., 2020). REG focuses on locating specific elements on the screen based on explicit references in the language instructions, such as "click the `Open` button." Collecting REG data from webpages is straightforward through crawling and parsing (Cheng et al., 2024; Chen et al., 2024b). However, when it comes to other platforms (e.g., desktop and mobile), it presents significant challenges and often requires substantial human effort.

Compared to REG, IG data is more crucial for real-world applications. IG can be considered a superset of REG, as it also includes actions that do not require specific coordinates, such as "Type". Moreover, the instructions in IG data are often nuanced and lack explicit element identification. For instance, an instruction like "delete the last file" requires reasoning to identify the targeted action type and element. IG data is often limited in size and diversity (Zhang et al., 2024b; Zheng et al., 2024b), due to the need for human annotation during collection (Li et al., 2024).

OS-Atlas tackles these data-related challenges by developing a multi-platform infrastructure for collecting GUI grounding data. A concurrent study by Gou et al. (2024) also addresses these challenges; however, their focus is limited to scaling web data.

## 3  OS-ATLAS

To establish a robust foundation action model for GUI agents, we propose enhancements from both data (§ 3.2) and methodological (§ 3.3) perspectives. Leveraging these innovations, we trained OS-Atlas, the first foundation action model specifically designed for GUI agents.

### 3.1  TASK FORMULATION AND TRAINING

Our training process consists of two consecutive phases: (1) GUI Grounding Pre-training, which equips VLMs with the knowledge to understand GUI screenshots and identify elements on the screen, and on top of it, (2) Action Fine-tuning, which transforms instructions into executable GUI actions. The framework overview can be found in Figure 2.

**GUI Grounding Pre-training.** This phase requires a large, high-quality, and diverse set of <screenshot, element referring expression or instruction, element coordinate> triplets, where the coordinates are represented as either points or bounding boxes. Models use the screenshot and the referring expression or instruction to predict the corresponding element coordinates. To facilitate large-scale pre-training, we have collected the largest multi-platform GUI reference corpus to date and synthesized a set of instruction grounding data using VLMs, as detailed in § 3.2. As shown in Table 1, our pre-training corpus covers 5 distinct platforms and includes over 2.3 million unique screenshots containing more than 13 million elements. We denote the pre-trained model as *OS-Atlas-Base*.

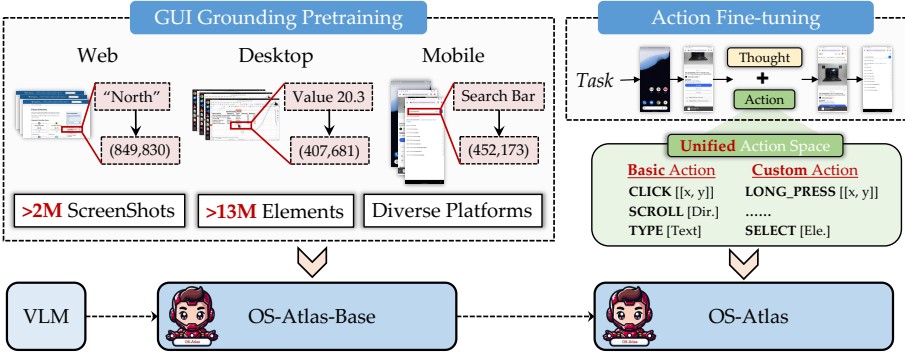

Figure 2: Overall training pipeline of OS-Atlas. We first perform large-scale pre-training using 13 million GUI grounding data collected to build OS-Atlas-Base. Next, we conduct multitask fine-tuning on agent data, resulting in OS-Atlas.

**Action Fine-tuning.**    To enable OS-Atlas to solve OS tasks effectively, we compile existing agent datasets for multi-task imitation learning. Specifically, we use <screenshot, task instruction, action history> triplets as model input and train the model to predict the corresponding action. Each action can be further represented as <thoughts, action type, action parameters(e.g., coordinates)> triplets. In our preliminary investigation, we discovered that fine-tuning with multiple diverse datasets can introduce conflicts between actions, which degrade performance (see § 5.3). To address this issue, we propose the use of a unified action space during training (see § 3.3).

## 3.2    GROUNDING DATA COLLECTION

As shown in Table 1, existing GUI grounding corpora predominantly focus on webpage screenshots, as these can be easily obtained using web crawlers (Cheng et al., 2024; Hong et al., 2024b; Chen et al., 2024b) or on mobile screenshots (You et al., 2024; Zhang et al., 2024d), leaving a significant gap for desktop screenshots. Furthermore, many of these corpora are either not open-sourced or are available only in relatively small scales. To lay a solid foundation for GUI agents, we have developed and open-sourced the first cross-platform GUI grounding data collection platform, along with a dataset comprising 13 million

| Dataset | #Screenshots | | | Open Source | #Elements |
| --- | --- | --- | --- | --- | --- |
| | Web | Mobile | Desktop | | |
| SeeClick | 270K | 94K | - | ✓ | 3.3M |
| Ferret-UI | - | 124K | - | ✗ | <1M |
| GUICourse | 73K | 9K | - | ✓ | 10.7M |
| CogAgent | 400K | - | - | ✗ | 70M |
| **OS-Atlas** | 1.9M | 285K | 54K | ✓ | 13.58M |

Table 1: Statistics of the grounding data we collected compared to existing efforts. (For open-source datasets, we only count the amount of data made publicly available.)

GUI grounding instances that cover Windows, macOS, Linux, Android, and the Web. However, due to significant discrepancies between these platforms, we were required to create distinct infrastructures for each one, which presents unique challenges in ensuring consistent data quality and compatibility across different environments.

**Web.**    We crawled about 4 million web pages from the latest URLs obtained from FineWeb (Penedo et al., 2024), a cleaned and deduplicated English dataset derived from CommonCrawl. For each webpage, we extracted all visible clickable elements from the HTML code — including buttons, scroll bars, search bars, hyperlinks, and SVG images with titles — along with their referring expressions and coordinates derived from the associated HTML attributes. Unlike previous methods (Cheng et al., 2024) that primarily focused on processing only the upper portions of websites, we render entire websites and then segment them into 1920x1080 resolution screenshots. This approach enhances the diversity of our web data by capturing a more comprehensive view of each webpage.

By excluding all error pages (e.g., 404 errors), we initially curated 3.7 million webpage screenshots and 37 million elements. However, upon human examination, we identified numerous low-quality

samples within this dataset. To address this issue, we implemented rule-based data filtering to exclude webpages that were either incompletely rendered or contained poorly distributed elements (e.g., all elements clustered at the bottom of the screen). Additionally, we restricted the maximum number of elements per webpage to 10 to encourage diversity. As a result of these stringent filtering criteria, we obtained a cleaned corpus consisting of 1.6 million screenshots and 7.7 million elements.

**Desktop & Mobile.** Capturing desktop and mobile screenshots is significantly more complex than collecting web screenshots. Previous methods primarily relied on manual collection, which resulted in a relatively small dataset. However, large-scale automated data collection presents the following challenges: (1) the substantial engineering efforts required to set up a simulation environment for data collection within a real operating system, and (2) the necessity of designing a program to mimic human interactions with the operating system, thereby changing system states to obtain new screenshots.

For Android, we utilize AndroidEnv (Toyama et al., 2021) to create a simulation environment, and for Linux, we employ OSWorld (Xie et al., 2024). Given the difficulties associated with virtualizing Windows and MacOS, we deploy the data synthesis platform on physical machines to collect data from these two operating systems. On these platforms, we leverage A11y tree to collect grounding data. Due to the differences in A11y tree APIs and tools supported by each operating system, we utilize *pyatspi* to access the A11y tree on Ubuntu, *pywinauto* on Windows, and *ApplicationServices* on macOS. We then simulate human-computer interactions by sampling actions from the obtained A11y tree. In our simulation environment, we employ two different exploration methods: Depth-First Search (DFS) and Random Walk. We apply a similar data filtering pipeline to the grounding data obtained as we did for the webpages.

**Instruction Grounding Data Collection.** In addition to the large-scale automated collection of referring expression data, we also annotated existing trajectory datasets using GPT-4o to obtain instruction grounding data. Given a high-level task instruction along with the before-and-after interface screenshots of an action, we instruct GPT-4o to carefully analyze the changes in the interface to derive a sub-instruction for the current action. Specifically, we employ Set-of-Mark prompting (Yang et al., 2023) to indicate the locations of the operated elements, which helps GPT-4o better comprehend the screenshots. We annotated the training sets of four trajectory datasets collected from both web and mobile platforms, namely Mind2Web (Deng et al., 2023b), AMEX (Chai et al., 2024), and AITZ (Zhang et al., 2024d). We also utilize instruction grounding data from two publicly available datasets: AndroidControl (Li et al., 2024) and Wave-UI [1].

### 3.3 UNIFIED ACTION SPACE

Our preliminary investigation found that blindly mixing data from different sources for multitask fine-tuning can significantly harm performance due to action space conflicts. For instance, the action "click" in a desktop environment is logically equivalent to the "tap" operation on a mobile device; training with such conflicts can confuse the model. To address this issue, we propose a unified action space that standardizes the format of all existing datasets. Our unified action space comprises both Basic Actions and Custom Actions. The prompt can be found in Table 6.

**Basic Actions.** These are standardized and available across all platforms. They provide essential functionality and are defined with a specific format, ensuring consistency and reliability. In the current design, we have three basic actions: click, type, and scroll. This design significantly reduces the size of action space when fine-tuning, and facilitates knowledge sharing across platforms and apps.

**Custom Actions.** These are unique to each user's platform and device. They enable the model to support new and unseen actions defined by users. The design of custom actions is crucial to OS-Atlas's good out-of-distribution performance, as they allow for on-demand extensions to support previously unseen tasks and actions. Typical custom actions include open_app (to open the specified application) and drag (to move an object to another location).

---

[1]https://huggingface.co/datasets/agentsea/wave-ui. We remove entries from ScreenSpot (Cheng et al., 2024), Mind2Web, and Omniact to avoid data contamination in downstream evaluation.

## 4 EXPERIMENTS: GROUNDING TASKS

### 4.1 EVALUATION DETAILS

**Benchmarks.** We begin by conducting a comprehensive evaluation of the GUI grounding performance of OS-Atlas-Base. Our evaluation utilizes ScreenSpot (Cheng et al., 2024), which assesses single-step GUI grounding capabilities across multiple platforms. During the evaluation, we identified approximately 11.32% percent annotation errors in the ScreenSpot dataset. To enhance the accuracy of our grounding evaluation, we corrected these errors and re-annotated certain examples, ensuring that the total number of test samples remains unchanged. In recognition of ScreenSpot's contributions, we have named the revised grounding dataset ScreenSpot-V2.

**Settings.** Following Gou et al. (2024), we evaluate under two settings: 1) *the Grounding Mode Setting*, which utilizes a planner model (e.g., gpt-4o) before grounding. The instructions from ScreenSpot are treated as subtask instructions and input into the planner to generate more detailed instructions for the grounding models. 2) *the Standard Setting* without a planner, which directly uses the original instructions from ScreenSpot.

**Models.** We consider two distinct backbone models: Qwen2-VL (Wang et al., 2024), which is trained explicitly with GUI data, and InternVL-2 (Chen et al., 2024d) which is trained without GUI data. These models also differ in their handling of image resolutions. InternVL-2-4B employs AnyRes (Liu et al., 2024; You et al., 2024) to resize images and segment larger images into smaller patches, which are then encoded independently using vision encoders. In contrast, Qwen2-VL-7B supports arbitrary image resolutions by directly mapping an image into a dynamic number of visual tokens. We denote our model as OS-Atlas-Base-4/7B, based on the backbones being used. Further details regarding the training setups can be found in Appendix E.

**Baselines.** We focus on VLMs that are explicitly trained with GUI data, including Fuyu (Bavishi et al., 2023), CogAgent (Hong et al., 2024b), Qwen2-VL (Wang et al., 2024), SeeClick (Cheng et al., 2024), and even a concurrent work UGround (Gou et al., 2024). We omit general VLMs such as GPT-4V, as they are well-studied and perform poorly on ScreenSpot (Cheng et al., 2024).

**Metrics.** We follow previous practices by using grounding accuracy on ScreenSpot, where a prediction is considered correct if the predicted location falls within the ground truth element's bounding box. However, this metric does not capture more fine-grained grounding errors. Therefore, we also use Intersection over Union (IoU), a widely used metric for measuring localization accuracy in object detection. IoU quantifies the overlap between the predicted bounding box and the ground truth bounding box.

### 4.2 RESULTS AND ANALYSIS

As shown in Table 2, under both settings, OS-Atlas-Base significantly outperforms previous grounding models on ScreenSpot across mobile, desktop, and web platforms, achieving state-of-the-art results. A similar trend is observed in ScreenSpot-V2 (see Appendix B). Notably, even for VLMs like Qwen2-VL, which have been pre-trained on GUI screenshots, incorporating GUI grounding pre-training can further enhance grounding capabilities. To gain deeper insights into the reasons behind this strong performance, we conducted a series of analyses under the standard setting (without a planner), including those in § 5.3, using InternVL-2-4B due to GPU constraints.

**The Effect of Grounding Data Scaling.** We plot the changes in grounding accuracy and IoU of OS-Atlas-Base-4B on ScreenSpot throughout the training process. As illustrated in Figure 3, grounding accuracy and IoU exhibit a clear positive correlation with the scaling of data, particularly in the case of IoU and the web domain, where we have nearly 10 million elements. The correlation is relatively weak in grounding accuracy because it cannot capture finer-grained errors. On one hand, this suggests the significant potential of continuously scaling the grounding data to further enhance performance. On the other hand, it underscores the need for more challenging benchmarks and improved metrics to effectively track performance improvements.

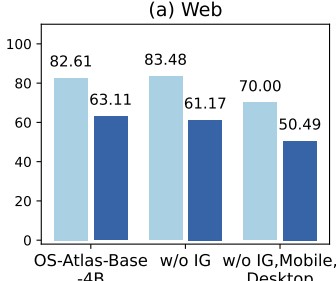 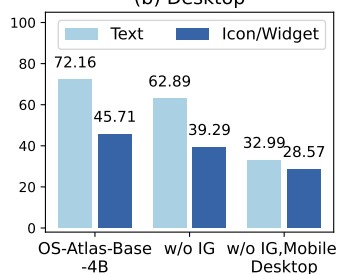 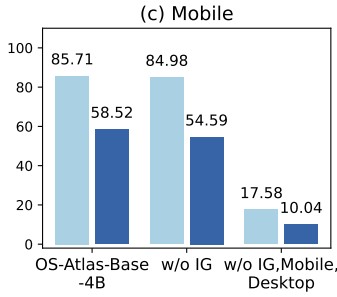

Figure 4: Ablation studies and performance on ScreenSpot. IG/Mobile/Desktop refers to instruction grounding, mobile, and desktop grounding data, respectively.

| Planner | Grounding Models | Mobile | | Desktop | | Web | | Avg. |
|---------|------------------|--------|-------------|---------|-------------|--------|-------------|------|
| | | Text | Icon/Widget | Text | Icon/Widget | Text | Icon/Widget | |
| - | Fuyu | 41.00 | 1.30 | 33.00 | 3.60 | 33.90 | 4.40 | 21.31 |
| | CogAgent | 67.00 | 24.00 | 74.20 | 20.00 | 70.40 | 28.60 | 49.58 |
| | SeeClick | 78.00 | 52.00 | 72.20 | 30.00 | 55.70 | 32.50 | 55.75 |
| | InternVL-2-4B | 9.16 | 4.80 | 4.64 | 4.29 | 0.87 | 0.10 | 4.32 |
| | Qwen2-VL-7B | 61.34 | 39.29 | 52.01 | 44.98 | 33.04 | 21.84 | 42.89 |
| | UGround-7B | 82.80 | 60.30 | 82.50 | 63.60 | 80.40 | 70.40 | 74.15 |
| | OS-Atlas-Base-4B | 85.71 | 58.52 | 72.16 | 45.71 | 82.61 | 63.11 | 70.13 |
| | OS-Atlas-Base-7B | **93.04** | **72.93** | **91.75** | **62.86** | **90.87** | **74.27** | **82.47** |
| GPT-4o | SeeClick | 83.52 | 59.39 | 82.47 | 35.00 | 66.96 | 35.44 | 62.89 |
| | UGround-7B | 93.40 | 76.90 | **92.80** | **67.90** | 88.70 | 68.90 | 82.71 |
| | OS-Atlas-Base-4B | **94.14** | 73.80 | 77.84 | 47.14 | 86.52 | 65.53 | 76.81 |
| | OS-Atlas-Base-7B | 93.77 | **79.91** | 90.21 | 66.43 | **92.61** | **79.13** | **85.14** |

Table 2: Grounding accuracy on ScreenSpot. The best results are in bold.

**Ablation.** We first remove the instruction grounding (IG) data from the pre-training phase to conduct a more controlled ablation analysis. Next, we further exclude mobile and desktop data to investigate whether pre-training solely on web data can generalize to other platforms. The results presented in Figure 4 reveal the following insights: (1) Referring expression data is nearly sufficient for training a strong grounding model and can be easily scaled compared to instruction grounding data. (2) Despite the similarities between different GUI platforms, pre-training solely on web data struggles to generalize to other platforms. This emphasizes the importance of our data infrastructure in facilitating the scaling of desktop and mobile referring expression data.

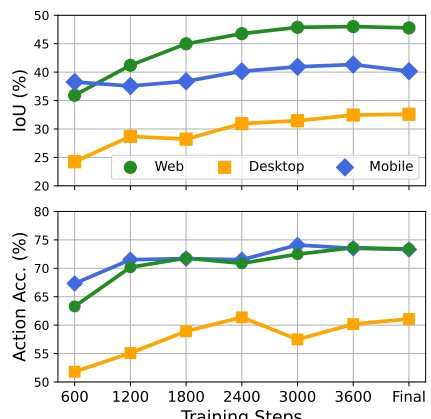

Figure 3: The effect of grounding data scaling on two metrics. The performances on three different domains are reported.

### 4.3 APPLICATION: GROUNDING MODE

We evaluate how OS-Atlas-Base work under the grounding mode in Figure 1: it can serve as a replacement for the grounding module of an existing GUI agent, thereby enhancing overall performance. In this study, we benchmark our approach on the challenging OS agent testbed, OSWorld (Xie et al., 2024). OSWorld is an interactive environment just like our computers, where the agent must

interact with the operating system at each step and wait for a response before proceeding to the next step. Refer to Figure 8 for a concrete example from the benchmark. Following their best practices, we constructed a screenshot-based GUI agent using GPT-4o. Given a specific task, the agent generates a detailed, step-by-step plan to accomplish it. It then executes this plan by generating actions and coordinates at each step. We substitute these coordinates with those generated by an external grounding model, either OS-Atlas-Base or SeeClick.

As shown in Table 3, although GPT-4o with OS-Atlas-Base as the grounding module still lags behind human performance, it significantly outperforms other grounding methods such as SeeClick and Set-of-Mark (SoM). This demonstrates the potential of OS-Atlas-Base as a standalone grounding module for developing future GUI agents.

| Models | Successful Rate | | | | | | | | | | Avg. |
|---|---|---|---|---|---|---|---|---|---|---|---|
| | OS | Calc | Impress | Writer | VLC | TB | Chrome | VSC | GIMP | WF | |
| GPT-4o + SoM | 20.83 | 0.00 | 6.77 | 4.35 | 6.53 | 0.00 | 4.35 | 4.35 | 0.00 | 3.60 | 4.59 |
| GPT-4o | 8.33 | 0.00 | 6.77 | 4.35 | 16.10 | 0.00 | 4.35 | 4.35 | 3.85 | 5.58 | 5.03 |
| + SeeClick | 16.67 | 0.00 | 12.76 | 4.35 | 23.52 | 6.67 | 10.86 | 8.70 | 11.54 | 7.92 | 9.21 |
| + OS-Atlas-Base-4B | 20.83 | 2.23 | 14.89 | 8.70 | 23.52 | 13.33 | 15.22 | 13.04 | 15.38 | 7.92 | 11.65 |
| + OS-Atlas-Base-7B | 25.00 | 4.26 | 17.02 | 8.70 | 29.41 | 26.67 | 19.57 | 17.39 | 19.23 | 8.91 | 14.63 |
| Human | 75.00 | 61.70 | 80.85 | 73.91 | 70.59 | 46.67 | 78.26 | 73.91 | 73.08 | 73.27 | 72.36 |

Table 3: Successful rate on OS World benchmark, divided by apps (domains). Workflow (WF) is a special domain that requires navigation across multiple apps.

## 5 EXPERIMENTS: AGENT TASKS

### 5.1 EXPERIMENT SETUPS

**Training details.** Given that there are currently relatively few agent benchmarks, especially in the desktop domain, we have only utilized three datasets — AMEX (Chai et al., 2024) (mobile), AITZ (Zhang et al., 2024d) (mobile), and Mind2Web (Deng et al., 2023a) (web) — to train our model, leaving a significant number of available benchmarks for OOD testing. For the sake of simplicity in notation, we denote our model as OS-Atlas-4/7B, which reflects the different backbone models utilized: InternVL-2-4B and Qwen2-VL-7B.

**Evaluation Benchmarks.** We examine five distinct agent benchmarks across three different platforms: AndroidControl (Li et al., 2024) and GUI-Odyssey (Lu et al., 2024a) for mobile agents; GUI-Act-Web (Chen et al., 2024a) and OmniAct-Web (Kapoor et al., 2024) for web agents; and OmniAct-Desktop for Windows environments. We only use the test split from these benchmarks for evaluation. Detailed statistics for these benchmarks can be found in Appendix C. Following common practices (Cheng et al., 2024; Deng et al., 2023a; Zhang et al., 2024d), we evaluate all benchmarks at the subtask granularity, as described in § 3.1. This involves allowing the model to predict actions for each step based on the task instruction, the associated screenshot, and action history (if available).

**Settings and Baselines.** We evaluate under two different settings to demonstrate two different practical applications of foundation action models like OS-Atlas: (1) zero-shot OOD setting (the *Action Mode* in Figure 1). In this setting, action models are benchmarked on unseen tasks, domains, or applications in a zero-shot manner, mimicking real-world usage scenarios for GUI agents.; (2) supervised fine-tuning setting (the *Agent Mode*): In this setting, researchers fine-tune models on downstream tasks to create agents specifically tailored for their intended applications.

In the zero-shot OOD setting, we use GPT-4o as the baseline, as existing VLMs perform poorly under this setting. For the supervised fine-tuning setting, we select InternVL-2, Qwen2-VL, and the grounding model, SeeClick, as our backbone for training.

**Metrics.** We evaluate our models using three commonly used metrics for GUI agents that assess the accuracy of action type prediction, coordinate prediction, and step success rate, denoted as *Type*, *Grounding*, and *SR*, respectively. *Type* measures the exact match score between the predicted action types (e.g., CLICK, SCROLL) and the ground truth, often referred to as Type EM in the literature.

| Models | GUI-Act-Web | | | OmniAct-Web | | | OmniAct-Desktop | | |
|---|---|---|---|---|---|---|---|---|---|
| | Type | Grounding | SR | Type | Grounding | SR | Type | Grounding | SR |
| Zero-shot OOD Setting | | | | | | | | | |
| GPT-4o | 77.09 | 45.02 | 41.84 | 79.33 | 42.79 | 34.06 | 79.97 | **63.25** | 50.67 |
| **OS-Atlas-4B** | 79.22 | 58.57 | 42.62 | 46.74 | 49.24 | 22.99 | 63.30 | 42.55 | 26.94 |
| **OS-Atlas-7B** | **86.95** | **75.61** | **57.02** | **86.12** | **69.35** | **59.99** | **90.24** | 62.87 | **56.73** |
| Supervised Fine-tuning Setting | | | | | | | | | |
| InternVL-2-4B | 81.42 | 47.03 | 36.17 | 47.51 | 51.34 | 24.39 | 67.00 | 44.47 | 29.80 |
| Qwen2-VL-7B | 89.36 | 90.66 | 82.27 | 89.22 | 85.94 | 78.58 | 96.27 | 94.52 | 91.77 |
| SeeClick | 88.79 | 78.59 | 72.34 | 86.98 | 75.48 | 68.59 | 96.79 | 70.22 | 72.69 |
| **OS-Atlas-4B** | **89.36** | 89.16 | 81.06 | 88.56 | 82.00 | 73.91 | 96.51 | 85.53 | 84.78 |
| **OS-Atlas-7B** | 89.08 | **91.60** | **82.70** | **97.15** | **95.41** | **93.56** | **97.15** | **95.85** | **94.05** |

Table 4: Results on web and desktop tasks. InternVL-2/Qwen2-VL and OS-Atlas-4/7B differ in that the former utilizes the original checkpoints, while the latter is fine-tuned on OS-Atlas-Base.

| Models | AndroidControl-Low | | | AndroidControl-High | | | GUI-Odyssey | | |
|---|---|---|---|---|---|---|---|---|---|
| | Type | Grounding | SR | Type | Grounding | SR | Type | Grounding | SR |
| Zero-shot OOD Setting | | | | | | | | | |
| GPT-4o | **74.33** | 38.67 | 28.39 | **63.06** | 30.90 | 21.17 | 37.50 | 14.17 | 5.36 |
| **OS-Atlas-4B** | 64.58 | 71.19 | 40.62 | 49.01 | 49.51 | 22.77 | 49.63 | 34.63 | 20.25 |
| **OS-Atlas-7B** | 73.00 | **73.37** | **50.94** | 57.44 | **54.90** | **29.83** | **60.42** | **39.74** | **26.96** |
| Supervised Fine-tuning Setting | | | | | | | | | |
| InternVL-2-4B | 90.94 | 84.05 | 80.10 | 84.09 | 72.73 | 66.72 | 82.13 | 55.53 | 51.45 |
| Qwen2-VL-7B | 91.94 | 86.50 | 82.56 | 83.83 | 77.68 | 69.72 | 83.54 | 65.89 | 60.23 |
| SeeClick | 93.00 | 73.42 | 75.00 | 82.94 | 62.87 | 59.11 | 70.99 | 52.44 | 53.92 |
| **OS-Atlas-4B** | 91.92 | 83.76 | 80.64 | 84.69 | 73.79 | 67.54 | 83.47 | 61.37 | 56.39 |
| **OS-Atlas-7B** | **93.61** | **87.97** | **85.22** | **85.22** | **78.48** | **71.17** | **84.47** | **67.80** | **61.98** |

Table 5: Results on mobile tasks. InternVL-2/Qwen2-VL and OS-Atlas-4/7B differ in that the former utilizes the original checkpoints, while the latter is fine-tuned on OS-Atlas-Base. AndroidControl-Low refers to the scenario where both low-level and high-level instructions are provided as inputs, while AndroidControl-High indicates that only high-level instructions are given.

*Grounding* evaluates the performance of GUI grounding in downstream tasks. *SR* represents the step-wise success rate, where a step is deemed successful only if both the predicted action and its associated arguments (e.g., coordinates for a click action) are correct. Appendix D provides detailed information on how these metrics are calculated.

## 5.2 RESULTS

The performances are presented in Table 4, 5. OS-Atlas achieved SOTA performance across three different platforms, six distinct datasets, and two evaluation settings. In comparison with GPT-4o, our model demonstrated superior capabilities in addressing unseen tasks across all six OOD evaluation datasets, even the desktop domain models haven't seen during action fine-tuning. This suggests that in the realm of GUI agents, OS-Atlas has the potential to be a robust open-source alternative to leading commercial VLMs. Additionally, the results of the SFT setting further confirm that OS-Atlas can serve as a robust foundation for researchers to train their custom GUI agents.

## 5.3 ANALYSIS

In this paper, we present two key research contributions: the development of a data infrastructure for grounding data synthesis and the proposal of a unified action space. We conduct experiments to analyze the significance of these factors in enhancing the zero-shot OOD performance of a foundational action model.

First, we investigate the effect of grounding pre-training by training OS-Atlas directly from the original VLMs, which we refer to as w/o pre-training. As illustrated in Figure 5, omitting the

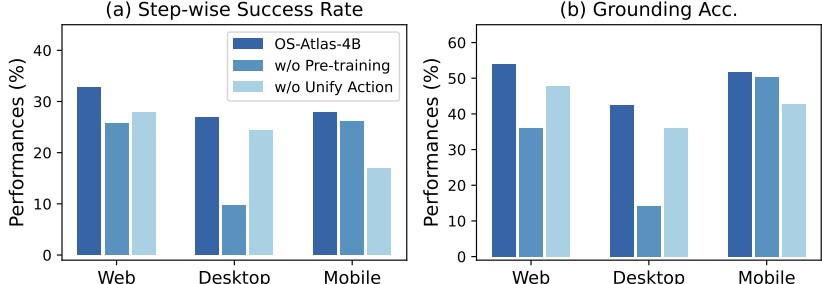

Figure 5: Ablation studies on the zero-shot OOD setting. The results are reported respectively across three platforms.

pre-training stage significantly degrades performance, particularly on the desktop and web platforms, where we have very limited data available for fine-tuning (7k samples for web and none for desktop). These results highlight the critical importance of the data infrastructure for grounding data synthesis; with this infrastructure in place, we can easily improve OOD downstream performance simply by scaling the pre-training corpus.

Next, we investigate the impact of removing the unified action space during fine-tuning, denoted as w/o unified action. For each fine-tuning dataset, we adhere to the optimal action space design proposed in SOTA models. As illustrated in Figure 5, we again observe a noticeable drop in performance. This validates our hypothesis that the conflicting action spaces indeed degrade model performance. Quantitatively, we find that employing our unified action space reduces the number of unique action types from 17 to 10, effectively resolving several naming conflicts, such as between "tap" and "click", "press_home" and "home", as well as "type" and "input".

## 5.4 OS-ATLAS-PRO

To ensure that most datasets remain available for OOD evaluation, OS-Atlas is initially trained using a limited selection of 3 agent datasets. To fully leverage its potential for broader applications, we use all 7 previously mentioned agent datasets for multitask fine-tuning. We report the average Success Rate (SR) across three domains: Web (GUI-Act-Web and OmniAct-Web), Mobile (AndroidControl-Low/High and GUI-Odyssey), and Desktop (OmniAct-Desktop). As illustrated in Figure 6, large-scale multitask fine-tuning significantly enhances model performance, thereby ensuring a better user experience when deployed in real-world applications.

## 6 CONCLUSION

In this paper, we present OS-Atlas, a foundation action model for GUI agents. OS-Atlas demonstrates exceptional performance in tackling open-environment GUI tasks across six complex benchmarks. This strong performance highlights the potential of OS-Atlas as an open-source alternative to powerful commercial VLMs, such as GPT-4o, for the development of future GUI agents.

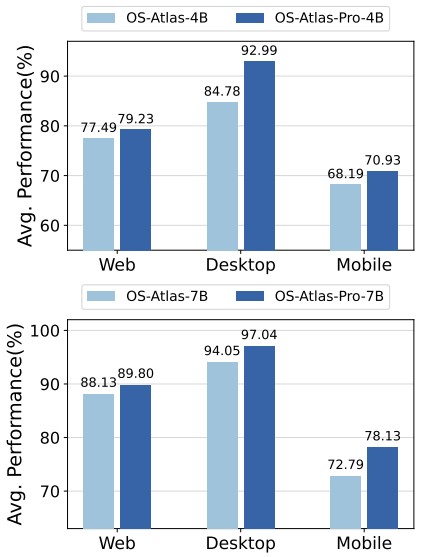

Figure 6: OS-Atlas-Pro evaluation results.

## ETHICS AND REPRODUCIBILITY STATEMENT

This research focuses on constructing a foundation action model for generalist GUI agents. The data used are obtained either from synthesizing or reprocessing from previously released datasets, with all datasets or benchmarks properly cited. There are no discrimination, bias, or fairness issues that need to be addressed in this paper. Further, our models are not expected to generate potentially harmful content. To ensure reproducibility, we provide all experimental details in Section 5 and their corresponding appendices. We will release all data, source code, and model checkpoints to support reproducibility.

## ACKNOWLEDGMENT

This work is supported by the Shanghai Artificial Intelligence Laboratory, with sincere appreciation for the support.

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

---

**Unified Action Space Prompt**

---

You are a foundational action model capable of automating tasks across various digital environments, including desktop systems like Windows, macOS, and Linux, as well as mobile platforms such as Android and iOS. You also excel in web browser environments. You will interact with digital devices in a human-like manner: by reading screenshots, analyzing them, and taking appropriate actions.

Your expertise covers two types of digital tasks:

  - **Grounding**: Given a screenshot and a description, you assist users in locating elements mentioned. Sometimes, you must infer which elements best fit the description when they aren't explicitly stated.

  - **Executable Language Grounding**: With a screenshot and task instruction, your goal is to determine the executable actions needed to complete the task. You should only respond with the Python code in the format as described below:

You are now operating in Executable Language Grounding mode. Your goal is to help users accomplish tasks by suggesting executable actions that best fit their needs. Your skill set includes both basic and custom actions:

**1. Basic Actions**

Basic actions are standardized and available across all platforms. They provide essential functionality and are defined with a specific format, ensuring consistency and reliability.

  Basic Action 1: CLICK
      - purpose: Click at the specified position.
      - format: CLICK <point>[[x-axis, y-axis]]</point>
      - example usage: CLICK <point>[[101, 872]]</point>
  Basic Action 2: TYPE
      - purpose: Enter specified text at the designated location.
      - format: TYPE [input text]
      - example usage: TYPE [Shanghai shopping mall]
  Basic Action 3: SCROLL
      - purpose: SCROLL in the specified direction.
      - format: SCROLL [direction (UP/DOWN/LEFT/RIGHT)]
      - example usage: SCROLL [UP]

**2.Custom Actions**

Custom actions are unique to each user's platform and environment. They allow for flexibility and adaptability, enabling the model to support new and unseen actions defined by users. These actions extend the functionality of the basic set, making the model more versatile and capable of handling specific tasks.

Your customized actions varied by datasets.

---

Table 6: The prompt for the action fine-tuning with a unified action space.

## A DATA STATISTICS

We detailed the statistics of the pre-training corpus we collected in Table 7.

## B SCREENSPOT-V2

During our error analysis of Screenspot, we identified that several errors stem from incorrect or ambiguous annotations in the benchmark. Specifically, we observed the following issues:

1. Some instructions contain spelling mistakes or reference elements that are not present in the screenshots.

2. Certain questions are ambiguous, allowing for multiple valid answers, while the ground truth includes only one of these options.

3. Several questions exhibit a high degree of similarity to one another.

| Training dataset | Type | Platform | Source | #Elements | #Screenshots |
|---|---|---|---|---|---|
| FineWeb-filtered | REG | Web | synthetic | 7,779,922 | 1,617,179 |
| Windows-desktop | REG | Windows | synthetic | 1,079,707 | 51,726 |
| Linux-desktop | REG | Linux | synthetic | 41,540 | 1,186 |
| MacOS-desktop | REG | MacOS | synthetic | 13,326 | 1,339 |
| Pixel6-mobile | REG | Mobile | synthetic | 104,598 | 21,745 |
| SeeClick | REG | Web & Mobile | public | 3,303,479 | 364,760 |
| AMEX | REG | Mobile | public | 1,097,691 | 99,939 |
| UIbert | REG | Mobile | public | 16660 | 5682 |
| Mind2Web-annotated | IG | Web | GPT-4o | 5,943 | 5,943 |
| AITZ-annotated | IG | Mobile | GPT-4o | 10,463 | 10,463 |
| AMEX-annotated | IG | Mobile | GPT-4o | 5,745 | 5,745 |
| AndroidControl | IG | Mobile | public | 47,658 | 47,658 |
| Wave-UI | IG | All platforms | public | 65,478 | 7,357 |
| **Total** | | | | **13,582,210** | **2,240,717** |

Table 7: Grounding training datasets statistics overview.

| Planner | Models | Mobile | | Desktop | | Web | | Avg. |
|---|---|---|---|---|---|---|---|---|
| | | Text | Icon/Widget | Text | Icon/Widget | Text | Icon/Widget | |
| - | SeeClick | 78.39 | 50.66 | 70.10 | 29.29 | 55.22 | 32.52 | 55.09 |
| | OS-Atlas-Base-4B | 87.24 | 59.72 | 72.68 | 46.43 | 85.90 | 63.05 | 71.86 |
| | OS-Atlas-Base-7B | **95.17** | **75.83** | **90.72** | **63.57** | **90.60** | **77.34** | **84.12** |
| GPT-4o | SeeClick | 85.17 | 58.77 | 79.90 | 37.14 | 72.65 | 30.05 | 63.60 |
| | OS-Atlas-Base-4B | 95.52 | 75.83 | 79.38 | 49.29 | 90.17 | 66.50 | 79.09 |
| | OS-Atlas-Base-7B | **96.21** | **83.41** | **89.69** | **69.29** | **94.02** | **79.80** | **87.11** |

Table 8: Grounding accuracy on ScreenSpot-v2. The best results are in bold.

4. Some ground truth bounding boxes are incorrectly labeled.

Given that the aforementioned factors could lead to biased evaluation results, we revised and edited the questions in the Screenspot benchmark. We ensured that the total number of questions remained the same in the release of Screenspot-v2. Our specific approach is outlined as follows:

1. We removed the problematic questions and replaced them with new ones.
2. We revised the instructions that were in the REG form and rewrote them as natural language instructions.
3. We corrected mislabeled ground truth bounding boxes.

We modified a total of 63 out of 436 ($\approx$14.4%) questions in the web domain, 28 out of 334 ($\approx$8.4%) in the desktop domain, and 53 out of 502 ($\approx$10.6%) in the mobile domain. The evaluation results on the new benchmark can be found in Table 8.

## C DETAILS OF EVALUATION BENCHMARKS

We display the statistical details of the evaluation benchmarks in Table 9. Notably, AndroidControl-Low denotes that both low-level and high-level instructions are provided as the inputs, while AndroidControl-High denotes that only the high-level instruction is in the input. Although screenshots from the training set of AndroidControl are used during the pretraining phase, we still classify it as an OOD dataset because it contains diverse OOD splits that differ from the training set. GUI-Odyssey-Random/Task/Device/App datasets are four different test splits based on the categories. We report the

macro-average performance across these splits. For OmniAct, the original dataset only provides the initial screenshot and does not have the dynamic environment, thus we evaluate the first action of each example under the OOD setting (Action mode). While under the supervised fine-tuning setting (Agent mode), we evaluate all actions in the trajectories.

| Benchmarks | Platforms | #Test Samples | History? | # Unified Actions |
|---|---|---|---|---|
| GUI-Act-Web | Web | 1,410 | | 3+2 |
| Omniact | Web | 1,427 | | 3+11 |
| | Desktop | 594 | | 3+11 |
| AndroidControl-Low | Mobile | 7,708 | ✓ | 3+5 |
| AndroidControl-High | Mobile | 7,708 | ✓ | 3+5 |
| GUI-Odyssey-Random | Mobile | 29,414 | | 3+6 |
| GUI-Odyssey-Task | Mobile | 17,920 | | 3+6 |
| GUI-Odyssey-Device | Mobile | 18,969 | | 3+6 |
| GUI-Odyssey-App | Mobile | 17,455 | | 3+6 |

Table 9: Details of the agentic benchmarks. *History* represents whether the history information of the previous actions is provided in the input. *#Unified Actions* denotes the number of actions (basic actions + custom actions) for each task.

## D   DETAILS OF EVALUATION METRICS

To ensure fair comparisons across all baseline methods, we standardize the evaluation metrics for each action.

For *click-based* actions (e.g., CLICK, LONG_PRESS), the action models must generate both the action type and the position coordinates (x,y). Since the ground-truth bounding box is not always available in the test data, we measure the performance by calculating the distance between the predicted coordinates and the ground-truth coordinates. Following Lu et al. (2024a), we consider the coordinates correct if they fall within a distance of 14% screen width from the ground truth.

*type-based* actions (e.g., TYPE, OPEN_APP) are considered correct if and only if both action type and action content are correct. We calculate the F1 score between the predicted text and the ground truth. The text is considered correct if F1 $>0.5$.

For *scroll* action, the direction argument (i.e., UP, DOWN, LEFT, and RIGHT) must precisely match the ground truth.

For other actions (e.g., PRESS_BACK), they must exactly match the ground truth to be considered correct.

## E   TRAINING DETAILS

**OS-Atlas-Base and OS-Atlas (4B)**   InternVL-2 employs Dynamic Aspect Ratio Matching to process dynamic high-resolution input. We set the `max_dynamic_patch` parameter to 6 to ensure the model captures sufficient pixel information. As a result, the input image, after resizing, is divided into a maximum of 6 tiles of 448×448 pixels, along with a thumbnail of the entire image to capture global context. In terms of grounding data format, to maintain consistency with the original InternVL training process, we convert all box format data into the form `<box>`[[x1, y1, x2, x2]]`</box>`, where (x1, y1) and (x2, y2) are the normalized relative coordinates within the range [0,1000]. Similarly, point data is converted into `<point>`[[x, y]]`</point>` format. `<box>`, `</box/>`, `<point>`, and `</point>` are treated as special tokens.

**OS-Atlas-Base and OS-Atlas (7B)**   Qwen2-VL can handle arbitrary image resolutions by mapping them into a dynamic number of visual tokens, offering a more human-like visual processing experience. Through our experiments, we discover that setting the `max_pixel` of image input to 1024x1024 during both training and inference yields excellent results for GUI grounding

tasks, while also optimizing the model's training and inference cost. Similarly, to maintain consistency with Qwen2-VL's original grounding training format, we convert box data into the format `<|box_start|>(x1,y1),(x2,y2)<|box_end|>`, where `<|box_start|>` and `<|box_end|>` are treated as special tokens.

We follow SeeClick to preprocess grounding pre-training data, formatting each REG data sample into three types: point grounding, box grounding, and OCR. Each type of data is wrapped up using 30 distinct GPT-generated prompts. To accelerate the training process, we group 15 samples into a single conversation, using 100 predefined prefix prompts.

## F  SCALING LAW OF GUI GROUNDING PRETRAINING

We also find that downstream task performance is not an ideal metric to measure scaling law. This is because downstream datasets often cannot accurately reflect the true data distribution, and evaluation metrics are too coarse-grained – for instance, correctly clicking an element does not necessarily mean the predicted coordinates exactly match the ground truth.

To study the scaling effect more rigorously, we plot the loss curve and, following Kaplan et al. (2020), fit a power law-based scaling curve, as shown in Figure 7. The horizontal axis represents the number of model training steps. Each step encompasses 1,024 samples, with each sample containing up to 15 grounding elements. The dark blue curve depicts the smoothed loss trajectory. From the figure, we observe a compelling trend suggesting significant potential for continually scaling pretraining data. Through our scaling law analysis, we estimate that increasing training data by 8 times could lead to a 40% relative reduction in loss. Moreover, scaling data by 64 times might potentially yield a 57% relative decrease in loss.

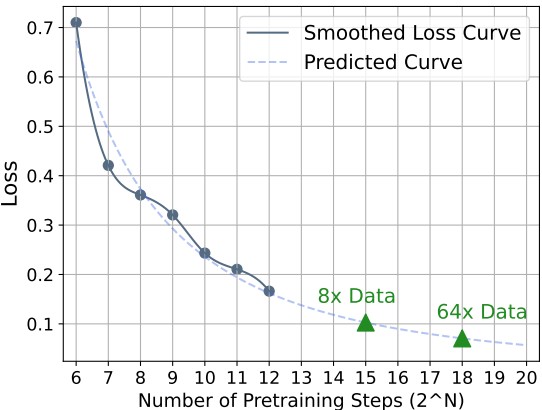

Figure 7: The curve of scaling law for the GUI grounding pretraining stage.

## G  CASE STUDY: OS-WORLD

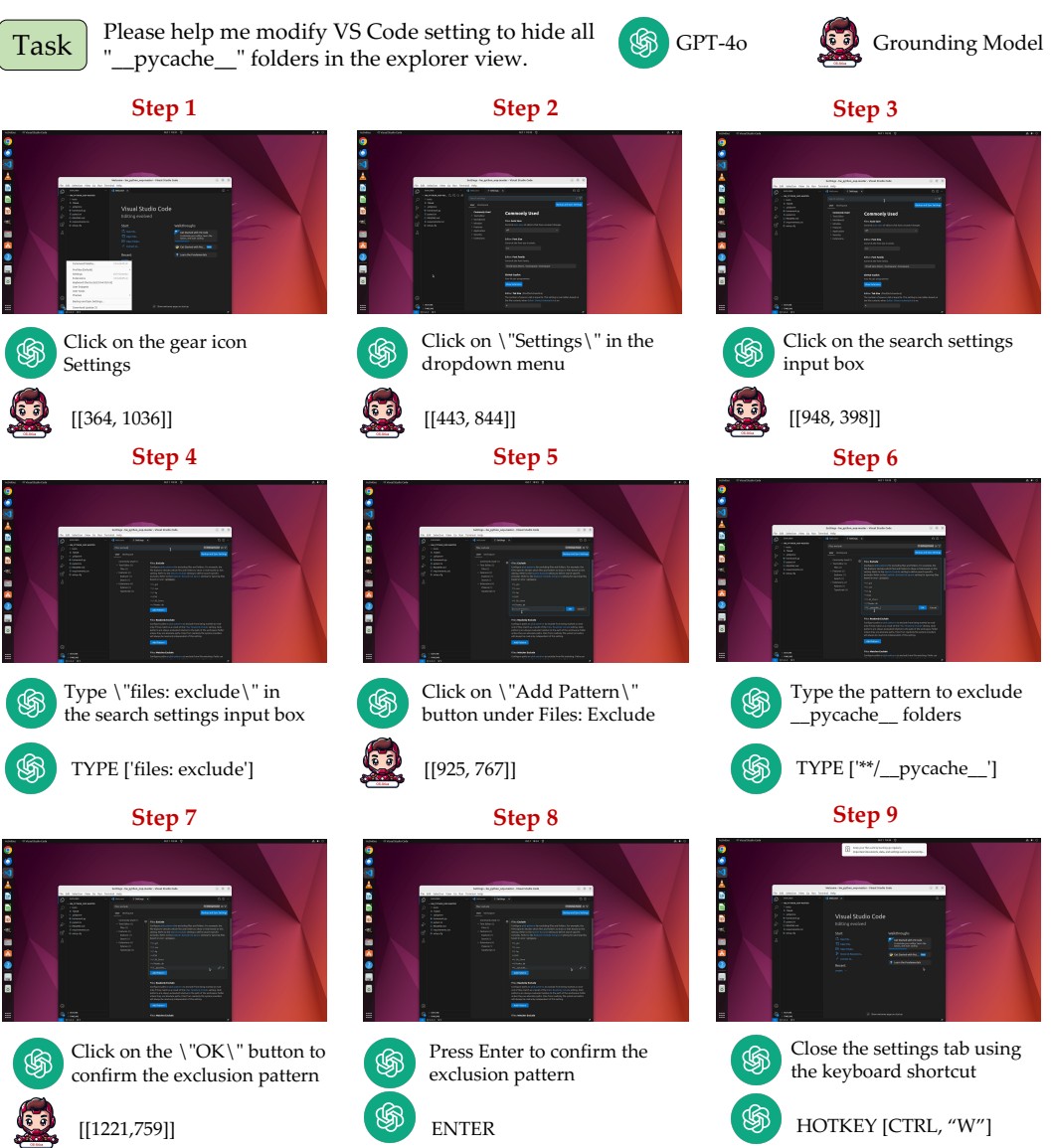

Figure 8: A case study from OS-World. OS-Atlas-Base works in the grounding mode, integrating GPT-4o as a task planner to create an agent. For each *Click* step, OS-Atlas-Base outputs the coordinates based on the provided step-level instructions.

