# OpenReview forum: "OS-ATLAS: Foundation Action Model for Generalist GUI Agents"
_ICLR.cc/2025/Conference — ICLR 2025 Spotlight_

### Official Review · Reviewer_giX8 · 2024-10-31

**Soundness:** 3
**Presentation:** 3
**Contribution:** 3
**Rating:** 8
**Confidence:** 4

**Summary:**

This paper proposes a high-performance GUI Agent, OS-ATLAS, trained on a synthetic grounding dataset and operating within a unified action space. Experiments on grounding tasks and agent tasks demonstrate that OS-ATLAS significantly outperforms previous state-of-the-art models.

**Strengths:**

This paper produces a large-scale, open-source corpus of screenshots covering multiple platforms and proposes a high-performance GUI Agent that significantly outperforms previous VLM-based GUI action models, making a substantial contribution to the field of vision-based GUI Agents. Additionally, the paper is logically coherent, relatively complete in content, and well-organized.

**Weaknesses:**

Experimental Concerns

1. Table 4 and Table 5 lack the results of state-of-the-art methods. It would be more appropriate to include these SOTA results for comparison with OS-ATLAS to demonstrate that it outperforms previous methods. Specifically, OdysseyAgent achieves 65.22% AMS (or SR, as referred to in this paper) on the GUI-Odyssey benchmark, which outperforms OS-ATLAS (4B/8B).

2. OS-ATLAS (8B) performs worse than OS-ATLAS (4B) on the AndroidControl-Low/High and GUI-Odyssey benchmarks. Moreover, OS-ATLAS (8B) exhibits poorer performance on the AndroidControl-High and GUI-Odyssey benchmarks compared to SeeClick, which uses Qwen-VL as the base model and a lower fixed input resolution (448x448). It would be beneficial to provide an explanation for this performance gap.

**Questions:**

Which benchmark is used in Figure 5?

---

> ### Author Response · Authors · 2024-11-21
> **Response part 1: Updated results in Table 4 and 5**
>
> Thanks for your questions. At the very beginning, we need to admit that we have a bug in the evaluation script of the 7B model. In particular, we forgot to skip certain special tokens when the 7B model is decoding. As a result, the 7B model's performance is much lower than expected. Note that this bug only happens to 7B models in Tables 4 and 5, but it does NOT influence other experiments. **Along with the release of our model weights, we will also release the eval scripts to help researchers reproduce those results.** You can find the new results in the updated paper. We also make a copy here for your convenience:
>
> > Results on Web and Desktop platforms under the OOD setting:
>
> |             |       | GUI-Act-Web |       |   |       | OmniAct-Web |       |   |       | OmniAct-Desktop |       |
> |-------------|:-------:|:-------------:|:-------:|---|:-------:|:-------------:|:-------:|---|:-------:|:-----------------:|:-------:|
> |             | Type  | Grd.        | SR    |   | Type  | Grd.        | SR    |   | Type  | Grd.            | SR    |
> | GPT-4o      | 77.09 | 45.02       | 41.84 |   | 79.33 | 42.79       | 34.06 |   | 79.97 | **63.25**           | 50.67 |
> | OS-Atlas-4B | 79.22 | 58.57       | 42.62 |   | 46.74 | 49.24       | 22.99 |   | 63.30 | 42.55           | 26.94 |
> | OS-Atlas-7B | **86.95** | **75.61**       | **57.02** |   | **86.12** | **69.35**       | **59.99** |   | **90.24** | 62.87           | **56.73** |
>
>
> > Results on Web and Desktop platforms under the supervised finetuning setting:
>
> |               |       | GUI-Act-Web |       |   |       | OmniAct-Web |        |        |   | OmniAct-Desktop |       |
> |---------------|:-------------:|:------:|:-------------:|------------|:-------------:|:------:|:-------------:|------------|:-------------:|:------:|:-------------:|
> |               | Type  | Grd.   | SR   |    | Type  | Grd.   | SR     |   |Type  | Grd. | SR    |
> | InternVL-2-4B | 81.42 | 47.03 | 36.17 |   | 47.51 | 51.34 | 24.39  |   | 67.00 | 44.47 | 29.80 |
> | Qwen2-VL-7B   | 89.36 | 90.66 | 82.27 |   | 89.22 | 85.94 | 78.58  |   | 96.27  | 94.52 | 91.77 |
> | SeeClick      | 88.79 | 78.59 | 72.34 |   | 86.98 | 75.48 | 68.59  |   | 96.79  | 70.22 | 72.69 |
> | OS-Atlas-4B   | **89.36** | 89.16 | 81.06 |   | 88.56 | 82.00 | 73.91  |   | 96.51  | 85.53 | 84.78 |
> | OS-Atlas-7B   | 89.08 | **91.60** | **82.70** |   | **97.15** | **95.41** | **93.56**  |   | **97.15**  | **95.85** | **94.05** |
>
> > Results on Mobile platform under the OOD setting:
>
> |             |       | AC-Low |       |   |       | AC-High |       |   |       | Odyssey |       |
> |-------------|:-------:|:--------:|:-------:|---|:-------:|:---------:|:-------:|---|:-------:|:---------:|:-------:|
> |             | Type  | Grd.   | SR    |   | Type  | Grd.    | SR    |   | Type  | Grd.    | SR    |
> | GPT-4o      | **74.33** | 38.67  | 28.39 |   | **63.06** | 30.90   | 21.17 |   | 37.50 | 14.17   | 5.36  |
> | OS-Atlas-4B | 64.58 | 71.19  | 40.62 |   | 49.01 | 49.51   | 22.77 |   | 49.63 | 34.63   | 20.25 |
> | OS-Atlas-7B | 73.00 | **73.37**  | **50.94** |   | 57.44 | **54.90**   | **29.83** |   | **60.42** | **39.74**   | **26.96** |
>
> > Results on Mobile platform under the supervised finetuning setting:
>
> |               |       | AC-Low |       |   |       | AC-High |        |        |   | Odyssey |       |
> |---------------|:-------------:|:------:|:-------------:|------------|:-------------:|:------:|:-------------:|------------|:-------------:|:------:|:-------------:|
> |               | Type  | Grd.   | SR   |    | Type  | Grd.   | SR     |   |Type  | Grd. | SR    |
> | InternVL-2-4B | 90.94 | 84.05 | 80.10 |   | 84.09 | 72.73 | 66.72  |   | 82.13 | 55.53 | 51.45 |
> | Qwen2-VL-7B   | 91.94 | 86.50 | 82.56 |   | 83.83 | 77.68 | 69.72  |   | 83.54  |  65.89 |  60.23 |
> | SeeClick      | 93.00 | 73.42 | 75.00 |   | 82.94 | 62.87 | 59.11  |   | 70.99  | 52.44 | 53.92 |
> | OS-Atlas-4B   | 91.92 | 83.76 | 80.64 |   | 84.69 | 73.79 | 67.54  |   | 83.47  | 61.37 | 56.39 |
> | OS-Atlas-7B   | **93.61** | **87.97** | **85.22** |   | **85.22** | **78.48** | **71.17**  |   | **84.47**  | **67.80** | **61.98** |

---

> > ### Author Response · Authors · 2024-11-21
> > **Response part 2**
> >
> > > Q1: Table 4 and Table 5 lack the results of state-of-the-art methods. It would be more appropriate to include these SOTA results for comparison with OS-ATLAS to demonstrate that it outperforms previous methods. Specifically, OdysseyAgent achieves 65.22% AMS (or SR, as referred to in this paper) on the GUI-Odyssey benchmark, which outperforms OS-ATLAS (4B/7B).
> >
> > Thanks for your suggestion. In the beginning, we omit the results of SOTA models because each benchmark has very different SOTA models, which will mess up the table. However, we agree that including SOTA model comparisons would provide a more comprehensive perspective. Below, we present the per-task success rate (SR):
> >
> > |                 | Gui-Act-Web | OmniAct-Web | OmniAct-Desktop | AC-Low  | AC-High  | GUI-Odyssey   |
> > |-----------------|:-----------:|:-----------:|:---------------:|:-------:|:--------:|:-------------:|
> > | SeeClick        | 72.34       | 68.59       | 72.69           | 75.00   | 59.11    | 53.92         |
> > | MiniCPM-GUI[1]  | 74.90       | -           | -               | -       | -        | -             |
> > | PaLM-2S[2]      | -           | -           | -               | 83.40   | 63.60    | -             |
> > | OdysseyAgent[3] | -           | -           | -               | -       | -        | 53.68 / 65.22 |
> > | OS-Atlas-4B     | 81.06       | 73.91       | 84.78           | 80.64   | 67.54    | 56.39 / 63.10 |
> > | OS-Atlas-7B     | **82.70**   | 93.56       | 94.05           |**85.22**| 71.17    | 61.98 / 68.16 |
> > | OS-Atlas-4B-Pro | 77.80       | 80.65       | 92.99           | 81.02   | 65.46    | 66.30         |
> > | OS-Atlas-7B-Pro | 80.99       | **98.60**   | **97.04**       | 84.29   | **71.55**| **78.54**     |
> >
> > We include SOTA models from each benchmark paper, such as MiniCPM-GUI (GUI-Act), PaLM-2S (AndroidControl), and OdysseyAgent (GUI-Odyssey). Since the OmniAct paper only evaluated open-source models with poor performance, we provide SeeClick results as its SOTA. As shown in the table above, OS-Atlas outperforms state-of-the-art models on Gui-Act, Omni-Act, and AC. Notably, OdysseyAgent reports results both with and without history input (53.68 and 65.22, respectively), where history represents a sequence of past actions. In contrast, most other datasets use past steps' instructions as history. Thus, in the submitted version, we did not utilize the history field from GUI-Odyssey during both training and inference. To ensure a fair comparison, we conducted additional experiments incorporating the action history input, which resulted in 63.10 and 68.16 for OS-Atlas-4B and 7B respectively. Under both settings, we found that OS-Atlas significantly outperforms OdysseyAgent.
> >
> > To ensure that most datasets remain available for OOD evaluation, OS-Atlas is initially trained using a limited
> > selection of 3 agent datasets. To fully leverage its potential for broader applications, we use all 7 agent datasets for multitask fine-tuning. We found that OS-Atlas-Pro generally outperforms our previous model across most datasets, with the exceptions of GUI-Act-Web and AndroidControl-Low. These two datasets have significantly larger training sets, which makes simple SFT prone to overfitting and results in artificially high performance.
> >
> > ---
> >
> > > Q2: OS-ATLAS (7B) performs worse than OS-ATLAS (4B) on the AndroidControl-Low/High and GUI-Odyssey benchmarks. Moreover, OS-ATLAS (7B) exhibits poorer performance on the AndroidControl-High and GUI-Odyssey benchmarks compared to SeeClick, which uses Qwen-VL as the base model and a lower fixed input resolution (448x448). It would be beneficial to provide an explanation for this performance gap.
> >
> > A2: As explained in Response Part 1 above, these are due to a mistake in the 7B models' evaluation scripts. With the updated Table 4 and 5, where OS-ATLAS (7B) now demonstrates significant performance improvements over both OS-ATLAS (4B) and SeeClick, your concerns shall be addressed.
> >
> > ---
> >
> > > Q3: Which benchmark is used in Figure 5?
> >
> > A3: We use benchmarks as in Table 4 and 5. In Figure 5, we report the averaged performances on each platform (e.g., Web domain includes Gui-Act-Web and OmniAct-Web). We will provide clearer explanations about the benchmark composition in a later version of the paper.

---

> > > ### Comment · Reviewer_giX8 · 2024-11-21
> > >
> > > Thanks for your explanation. The updated results appear to be sound. With the experimental setup properly aligned, OS-Atlas demonstrates SOTA performance. I have no further questions.

---

> > > > ### Author Response · Authors · 2024-11-22
> > > >
> > > > Thanks so much for taking a closer look at our rebuttal. We really appreciate how carefully you considered our responses and the willingness to reconsider the paper's score. Your thoughtful feedback has been incredibly helpful in refining our research.

---

### Official Review · Reviewer_byA6 · 2024-11-02

**Soundness:** 3
**Presentation:** 3
**Contribution:** 3
**Rating:** 8
**Confidence:** 4

**Summary:**

The paper aims to enhance foundation model for GUI agent research. The authors introduce a GUI grounding synthesis tool and collect a large-scale dataset using the toolkit. Using the datasets, OS-Atlas is developed, which is a cross-platform foundation action model for GUI. Detailed results and evaluations are also discussed.

**Strengths:**

- Open-source toolkit and data are available for the research community, which is currently lacking in GUI agent research area
- Data collection and modeling processes are extensive and well-planned and motivated.

**Weaknesses:**

- I would appreciate more details about the toolkit, particularly on its usability and how other researchers could potentially customize it for their own use cases.

**Questions:**

- Could the authors comment on the usability of the toolkit and how other researchers could potentially customize it for their own use cases.
- Any discussions on choosing a pertaining + fine-tuning over mixing grounding and action data?

---

> ### Author Response · Authors · 2024-11-21
>
> > Q1: I would appreciate more details about the toolkit, particularly on its usability and how other researchers could potentially customize it for their own use cases.
>
> A1: We will provide detailed documentation alongside the release of the toolkit. In brief, our goal is to make the toolkit as user-friendly as possible, allowing users to simply click and go.
>
> - The web toolkit is the easiest to use. Researchers only need to prepare a list of web URLs they wish to crawl and then initiate the synthesis toolkit with a single click.
> - For mobile and desktop toolkits, researchers must manually install the applications from which they want to collect data. They will also need to adjust the synthesis configuration accordingly before starting the script. Once initiated, the script will automate the remaining data synthesis process.
>
> Additionally, we will include tutorials that demonstrate how to modify the scripts to collect specific data types (e.g., for those interested in gathering more grounding data for a search bar).
>
> ---
>
> > Q2: Any discussions on choosing a pertaining + fine-tuning over mixing grounding and action data?
>
> A2: Thank you for the insightful question. First, we must acknowledge that we chose a two-stage training approach primarily due to GPU constraints. Given the large number of SFT experiments we conducted, which involved fine-tuning the pretrained grounding model on a single agent dataset, we could not afford to repeatedly retrain the model using grounding data combined with different action data splits.
>
> Following your suggestion, we experimented with a mixed-training strategy that combines all agentic data from six benchmarks with all grounding data. The performance of the mixed-trained model is comparable to that of the two-stage training strategy. Interestingly, we found that the mixed-trained model is more robust to changes in the prompts. Due to the short rebuttal time and GPU constraints, we are experimenting with a 1B model now. But we will scale the experiments to 4B/7B, and add a section in the appendix to discuss those findings.

---

> > ### Comment · Reviewer_byA6 · 2024-12-02
> >
> > Thank you for your response, I have no further questions.

---

### Official Review · Reviewer_Gb96 · 2024-11-03

**Soundness:** 3
**Presentation:** 3
**Contribution:** 3
**Rating:** 8
**Confidence:** 4

**Summary:**

This paper presents a project that aims to replace commercial VLMs such as GPT-4o in serving as the base model for GUI agents, with a special focus on GUI grounding and atomic action execution. The project involves a large scale grounding and action dataset, and a trained model named OS-Atlas. Benchmarking on various grounding and atomic action datasets show SOTA performance.

**Strengths:**

The main claim of the paper is to serve as an open-source alternative to commercial VLMs like GPT-4o in building GUI agents. The claim is well supported, with the condition that all the data, code, and model are open-sourced as promised.

Specifically, the contribution mainly involves (1) a dataset with 2.3 million screenshots on various platforms and operating systems; (2) OS-Atlas (4B/8B) model that achieves SOTA performance on GUI grounding and atomic action execution.

Having a good base model and data would be beneficial for the community on future GUI research.

**Weaknesses:**

1. The contribution and novelty limited to the main claim of serving as an open-source alternative to commercial VLMs. Such VLMs like GPT-4o are not designed specifically for GUI scenarios, and fail to address GUI specific challenges. So does this work.

1.1. The study does not touch any GUI specific challenges, such as specific visual and text distribution, cross-system differences, long action planning, and so on.

1.2. It also does not present new evaluations that are more suitable for GUI. For example, setting up an emulator-style benchmark with the data synthesis toolkit.

Nonetheless, those aspects are beyond the claim of the paper, and future research on related topics could benefit from the data and model released by this study.

2. Extra analyses would be beneficial:

2.1. There could be more analyses on the data and model scaling, to help future study understand the amount of extra data and compute needed if the application needs further model enhancement or finetuning. The existing scaling analyses in Figure 3 and Sec. 4.2. are somewhat noisy and fail to give clear directions.

2.2. To better support the claim on being effective “universally across all GUIs,” there could be more discussion on the per-task and environment performance. It remains unclear why certain tasks perform better or worse on web/desktop/mobile and different operating systems. Do such differences attribute to model, data, or task setting? And what are the corresponding data and compute needed to improve on a specific setting of interest.

**Questions:**

Please see weaknesses for questions, mostly on the analyses part. The released data and model would be a good contribution to the community, looking forward to seeing it being released.

---

> ### Author Response · Authors · 2024-11-21
>
> > Q1: The study does not touch any GUI specific challenges, such as specific visual and text distribution, cross-system differences, long action planning, and so on.  It also does not present new evaluations that are more suitable for GUI. For example, setting up an emulator-style benchmark with the data synthesis toolkit. Nonetheless, those aspects are beyond the claim of the paper, and future research on related topics could benefit from the data and model released by this study.
>
> A1: We believe we have made efforts to address some key challenges in GUI interaction: GUI element grounding, single-step GUI problem solving (i.e., action grounding), and their cross-system generalization. We have conducted evaluations on OSWorld (Table 3), an emulator-style interactive benchmark. However, we agree with the reviewer that additional GUI-specific challenges, such as long-term action planning and developing more comprehensive benchmarks, are important directions that need to be addressed in future research.
>
> ---
>
> > Q2: There could be more analyses on the data and model scaling, to help future study understand the amount of extra data and compute needed if the application needs further model enhancement or finetuning. The existing scaling analyses in Figure 3 and Sec. 4.2. are somewhat noisy and fail to give clear directions.
>
> A2: Thanks for pointing this out. We also find that downstream task performance is not an ideal metric to measure scaling law and can be noisy. This is because downstream datasets often cannot accurately reflect the true data distribution, and evaluation metrics are too coarse-grained – for instance, correctly clicking an element does not necessarily mean the predicted coordinates exactly match the ground truth.
>
> To study the scaling effect more rigorously, we plot the loss curve and, following [1], fit a power law-based scaling curve, as shown in Figure 7 in the updated paper's appendix. Through our scaling law analysis, we estimate that increasing training data by 8 times could lead to a 40% relative reduction in loss. Moreover, scaling data by 64 times might potentially yield a 57% relative decrease in loss.
>
> [1] Scaling laws for neural language models.
>
> ---
>
> > Q3: To better support the claim on being effective “universally across all GUIs,” there could be more discussion on the per-task and environment performance. It remains unclear why certain tasks perform better or worse on web/desktop/mobile and different operating systems.
>
> A3: The primary reason behind varying performance across datasets is their inherent difficulty levels. For instance, in the desktop domain, we achieve over 90% success rate (SR) on OmniAct, while for OSWorld, even human performance is below 80%. Consequently, comparing absolute performance across benchmarks is not particularly insightful. However, we find that desktop tasks are generally more complex than mobile and web domain tasks, due to the inherent complexity of desktop screenshots.
>
> Another reason behind varying performance across datasets is that different benchmarks employ varied experimental setups. To maintain our model's universality, we intentionally avoided over-optimizing our design for specific benchmarks. Instead, we adopted a unified evaluation setting to demonstrate our model's generalizability across GUI environments.
> Take GUI-Odyssey as an example: our best model, OS-Atlas-7B, initially achieves a 61% SR as in Table 5, compared to their SOTA model OdysseyAgent at 65% SR. The performance difference stems from their unique approach of using previous action sequences as historical context to inform predictions. By adopting their proposed best practice, we can improve our performance to 68% SR.
>
> In essence, achieving optimal performance on certain tasks would require case-by-case considerations regarding model, data, and computational scaling, which is not the focus of our paper. But we are more than willing to continue this discussion to share our observations and learned lessons if you are interested in certain benchmarks.

---

> > ### Comment · Reviewer_Gb96 · 2024-11-29
> > **Thank you to the authors for the detailed response.**
> >
> > I will raise my score to an accept, as most of my other concerns have been addressed.

---

> > > ### Author Response · Authors · 2024-11-29
> > >
> > > Thanks for your constructive question and strong support. Please feel free to open the discussion if you have further questions.

---

### Official Review · Reviewer_eJ95 · 2024-11-05

**Soundness:** 3
**Presentation:** 2
**Contribution:** 3
**Rating:** 6
**Confidence:** 4

**Summary:**

The work introduces OS-ATLAS, a foundational GUI action model for GUI grounding and Out-Of-Distribution (OOD) agentic tasks. The authors synthesize large-scale multi-platform GUI grounding data. The OS-ATLAS model trains on this data and further fine-tunes with action datasets to address specific agent tasks. The model demonstrates significant performance improvements over previous models and shows the effectiveness of the synthetic GUI grounding dataset. The author also promises to open-source the data and related models, which will significantly impact the development of the GUI agent.

**Strengths:**

1. The work proposes a pretraining-then-fine-tuning framework for GUI grounding and agentic tasks, which is natural and unifies the training and evaluation process of these tasks.
2. The work synthesizes a large-scale multi-platform GUI grounding data for pertaining and promises to open-source the data.
3. The experimental results showed significant improvement, demonstrating their effectiveness.

**Weaknesses:**

1. The authors propose synthesizing large-scale GUI grounding datasets across platforms. However, the number of instances in Table 1 is not similar for each platform, especially for desktop (54K).
2. There are some missing details: how to ensure the correctness of the synthetic data, the specifics of action tuning and the format of the inputs and outputs, and the precise meaning of "thought."

Note that, the score given is based on all concerns addressed in the discussion stage.

**Questions:**

See the Weakness part.

---

> ### Author Response · Authors · 2024-11-21
>
> > Q1: The number of instances in Table 1 is not similar for each platform, especially for desktop (54K)
>
> A1:  The considerable difference in the volume of web data compared to desktop data arises from the inherent diversity of the data itself. Statistics show that the average person uses 9 mobile apps daily and 30 apps monthly in 2024.[1] In contrast, the average internet user has been visiting around 30 unique web pages each day in 2010.[2] As a result, the diversity of desktop data is significantly lower than that of web data.
>
> Our dataset of 54K desktop entries includes system apps and the 10 most commonly used third-party apps. In contrast, our web data comes from millions of different websites. We initially gathered 330K desktop data samples and applied strict filtering criteria to ensure a high-quality dataset.  The experimental results in ScreenSpot's desktop domain and the ablation analysis, validate the effectiveness of this data. Nonetheless, we are eager to collaborate with the community to further scale both desktop and mobile data following the release of our toolkit.
>
> [1] https://buildfire.com/app-statistics/
>
> [2] Differential Internet Behavior’s of Students from Gender Groups. IJCA 2010
>
> ---
>
> > Q2: There are some missing details: how to ensure the correctness of the synthetic data, the specifics of action tuning and the format of the inputs and outputs, and the precise meaning of "thought."
>
> A2:  Thanks for pointing this out, we will include a concrete example and the following clarification regarding action fine-tuning:
>
> ```python
> Inputs:
> <system_prompt>
> <image>
> Task: In the Arts & Culture app, I want to create an art gallery with the title Self Art.
> History:
> Step 1: Click on the Favorites
> Step 2: Click on the Galleries section
>
> Outputs:
> thoughts:
> Click on the Create Gallery button
> actions:
> CLICK <point>[[255,439]]</point>
> ```
> During action finetuning, as shown in the example, the model will take the following input:
> - A system prompt, as detailed in Table 6 in the appendix.
> - The task associated screenshot.
> - Task instruction and action histories, which contain thoughts or subtask instructions from previous steps.
>
> The model is trained to predict the following two fields:
> - thoughts:  the reasoning of what the model should do at the current step. You can also understand it as a model generated subtask instruction.
> - actions: action type and parameters (e.g., coordinates)
>
> ---
>
> > Q3: How to ensure the correctness of the synthetic data
>
> A3: We apply both rule-based and model-based methods to synthesize data. For rule-based methods, the data is considered correct as long as the human annotations in the annotated ally tree are accurate. We only need to filter out data with rendering errors or network errors, as detailed in lines 210-215.
>
> For model-synthesized data, validating correctness remains a challenge even the entire LLM community is struggling with. We can only assess the correctness and effectiveness of the data through the model's performance after training. During our preliminary investigation, we conducted experiments to ensure that incorporating these model-synthesized data would not degrade the model's performance. In one controlled evaluation, we found that after pre-training on rule-synthesized data, continuous training on model-synthesized data slightly improves performance.
>
> We also randomly sample 50 data entries from each split to manually validate their correctness before proceeding with final model training.

---

### Author Response · Authors · 2024-11-21
**General Response to all reviewers**

Dear ACs and Reviewers,

We sincerely appreciate the thoughtful and constructive feedback provided in your reviews. We have carefully addressed each of your concerns, and this response outlines the updates made to our manuscript.

- Regarding the model naming convention from the original Qwen2-VL paper, we have adjusted the model name from OS-Atlas-8B to OS-Atlas-7B, aligning with the original Qwen paper, despite the model containing 8.4B parameters.

- We have updated the results in Tables 4 and 5. Please refer to 'Response Part 1' to Reviewer giX8 for comprehensive explanations.

- To support broader use and future research, we have introduced the OS-Atlas-Pro models (4B / 7B) in section 5.4.

- According to the suggestion from Reviewer Gb96, we have added a discussion of the GUI grounding pretraining scaling law in Appendix F.

Best regards,

Authors of OS-Atlas

---

### Meta-Review · Area_Chair_dRth · 2024-12-17

**Metareview:**

The work introduces a new model for GUI grounding and UI interaction. The paper introduces a toolkit to curate synthetic GUI grounding data for multiple platforms including mobile, desktop and web platforms. Using this toolkit, they collect a large (2.3 million instances) dataset for multi-platform GUI grounding. A model trained on this data shows improvements across multiple platforms on the popular screenspot benchmark. Further, they also train the model to perform single-step navigation given the current screenshot, goal and action history and show improvements compared to past methods on navigation benchmarks for mobile, web and desktop.

All reviewers agree that the toolkit, associated data and models will be a useful resource to the community, and will help further research in building digital UI interaction models. All reviewers also agreed that the experiments conducted in the manuscript are sound and exhaustive, and the model trained on their data shows strong performance compared to previous baselines.

Overall, I think the contributions (data, models) of the paper will be useful to the community, especially, as the problem of digital task execution is catching the attention of the community in recent times.

**Additional Comments On Reviewer Discussion:**

The reviewer discussion period for the paper was productive and a few points stood out.

Reviewer eJ95g  raised concerns about ensuring correctness of the synthetic data. The authors performed some rule-based filtering to remove data with rendering or networking errors. The authors also noticed that incorporating synthetic data improves model's performance but agreed that ensuring correctness of synthetic data for such a large dataset is challenging. We encourage the authors to include this as limitations in the final manuscript.

Reviewer Gb96 asked several clarifying questions about scaling laws for data and models The authors performed some analysis during the rebuttal which should be added as part of the appendix. There was also some discussion during the reviewer discussion phase around measuring performance across environments which would be useful to present in the appendix.

Reviewer giX8 encouraged the authors to include previous SOTA methods for each environment. During the discussion phase, the authors found a bug in their evaluation, fixing which, the numbers improved. Providing a more exhaustive table and accurately representing state of the art methods in the appendix will make the paper self-sufficient and will improve the readability of the paper.

---

### Decision · Program_Chairs · 2025-01-22

Accept (Spotlight)